# The Prevalence of Human *Plasmodium* Species during Peak Transmission Seasons from 2016 to 2021 in the Rural Commune of Ntjiba, Mali

**DOI:** 10.3390/tropicalmed8090438

**Published:** 2023-09-07

**Authors:** Francois Dao, Laurent Dembele, Bakoroba Diarra, Fanta Sogore, Alejandro Marin-Menendez, Siaka Goita, Aboubacrin S. Haidara, Yacouba N. Barre, Cheick P. O. Sangare, Aminatou Kone, Dinkorma T. Ouologuem, Antoine Dara, Mamadou M. Tekete, Arthur M. Talman, Abdoulaye A. Djimde

**Affiliations:** 1Malaria Research and Training Center, Faculty of Pharmacy, Faculty of Medicine and Dentistry, University of Sciences, Techniques, and Technologies of Bamako, Bamako 1805, Mali; francoisd@icermali.org (F.D.); bakoroba.diarra@icermali.org (B.D.); fanta.sogore@icermali.org (F.S.); sgoita@icermali.org (S.G.); ahaidara@icermali.org (A.S.H.); ynbarre@icermali.org (Y.N.B.); oumar@icermali.org (C.P.O.S.); amina@icermali.org (A.K.); ina@icermali.org (D.T.O.); tonydara@icermali.org (A.D.); mtekete@icermali.org (M.M.T.); 2MIVEGEC, Université de Montpellier, CNRS, IRD, 34095 Montpellier, France; alejandro.marinmenendez@ird.fr (A.M.-M.); arthur.talman@ird.fr (A.M.T.)

**Keywords:** *Plasmodium*, epidemiology, transmission, seasonality

## Abstract

Up-to-date knowledge of key epidemiological aspects of each *Plasmodium* species is necessary for making informed decisions on targeted interventions and control strategies to eliminate each of them. This study aims to describe the epidemiology of plasmodial species in Mali, where malaria is hyperendemic and seasonal. Data reports collected during high-transmission season over six consecutive years were analyzed to summarize malaria epidemiology. Malaria species and density were from blood smear microscopy. Data from 6870 symptomatic and 1740 asymptomatic participants were analyzed. The median age of participants was 12 years, and the sex ratio (male/female) was 0.81. Malaria prevalence from all *Plasmodium* species was 65.20% (95% CI: 60.10–69.89%) and 22.41% (CI: 16.60–28.79%) for passive and active screening, respectively. *P. falciparum* was the most prevalent species encountered in active and passive screening (59.33%, 19.31%). This prevalence was followed by *P. malariae* (1.50%, 1.15%) and *P. ovale* (0.32%, 0.06%). Regarding frequency, *P. falciparum* was more frequent in symptomatic individuals (96.77% vs. 93.24%, *p* = 0.014). In contrast, *P. malariae* was more frequent in asymptomatic individuals (5.64% vs. 2.45%, *p* < 0.001). *P. ovale* remained the least frequent species (less than 1%), and no *P. vivax* was detected. The most frequent coinfections were *P. falciparum* and *P. malariae* (0.56%). Children aged 5–9 presented the highest frequency of *P. falciparum* infections (41.91%). Non-*falciparum* species were primarily detected in adolescents (10–14 years) with frequencies above 50%. Only *P. falciparum* infections had parasitemias greater than 100,000 parasites per µL of blood. *P. falciparum* gametocytes were found with variable prevalence across age groups. Our data highlight that *P. falciparum* represented the first burden, but other non-*falciparum* species were also important. Increasing attention to *P. malariae* and *P. ovale* is essential if malaria elimination is to be achieved.

## 1. Introduction

The World Health Organisation (WHO) estimated the number of malaria cases to be 247 million and the number of malaria deaths to be 619,000 worldwide in 2021. In 2021, 95% of deaths due to malaria were in Africa [1]. Over the past two decades, a downward trend was observed in the global incidence of malaria, dropping from 80 to 57 cases per 1000 people at risk from 2000 to 2019 [2]. Similarly, malaria-associated deaths declined from 25 to 10 deaths per 100,000 at-risk inhabitants [2]. However, in the sub-Saharan African region, the burden of malaria has remained unchanged [3] despite all of those endemic countries having adopted elimination strategies. Eleven of the world’s most malaria-prone countries, including Mali, failed to control the disease during the COVID-19 pandemic. Although malaria deaths in these countries decreased from 444,600 in 2020 to 427,854 in 2021, malaria cases increased from 165 million to 168 million during the same period [4]. While *P. falciparum* malaria declined, *P. malariae* and *P. ovale* infections rose from two- to sixfold [5].

*P. falciparum* and *P. vivax* are the most significant threats among the five human-causing malaria species. In 2018, *P. falciparum* accounted for 99.7% of estimated malaria cases in Africa, while *P. vivax* predominated in America, accounting for 75% of the cases [6]. *P. malariae was* the third most frequent species, with a prevalence of up to 15–40% and an overlapping distribution with *P. falciparum* throughout tropical Africa and malaria-endemic regions of the world [5,7,8]. *P. ovale* is predominantly found in forests and humid savannah areas in West and Central Africa, with a prevalence of 4 to 10% [9]. The fifth species, *P. knowlesi,* initially thought to infect nonhuman primates, was reported to also infect humans during the 2004 human malaria outbreak in Southeast Asia [10,11].

Mali is among the ten countries with the highest morbidity and mortality due to *Plasmodium* infections, contributing to 3% of malaria cases and deaths worldwide and 6% of cases in West Africa. Between 2016 and 2019, malaria cases decreased by 13%, from 384 to 334 per 1000 inhabitants [12]. Similarly, malaria-associated deaths decreased by 21%, from 0.76 to 0.59 for 1000 [12]. The heterogeneous transmission of malaria across Mali causes significant variations in malaria prevalence across the country, going from 1% in the capital health district of Bamako to 30% in the Sikasso region [13]. Despite multiple efforts deployed against malaria focused on vector control, the massive deployment of long-acting insecticide-treated mosquito nets (LLINs), intermittent preventive treatment (IPT), seasonal malaria chemoprevention (CPS), free policy for children under five and pregnant women, rapid diagnosis and cases management at the community level, malaria cases remain high in Mali. Of the six human malaria species, *P. falciparum*, *P. malariae*, *P. ovale*, and *P. vivax* are those causing malaria in Mali [14,15]. *P. falciparum,* the most prevalent species, causes the most lethal episodes, and depending on the locality, this species is responsible for 82 to 95% of all malaria cases [16,17]. As in many sub-Saharan Africa endemic countries, the burden of *P. malariae* and *P. ovale* is poorly understood, hampering disease elimination.

This study aims to fill the gap by describing the prevalence of plasmodial species encountered in a hyperendemic zone of Mali, Ntjiba, during high seasonal malaria transmission. The commune of Ntjiba, located in the Guinean Sudanese zone, is characterized by seasonal malaria transmission. Analyzing retrospective data from previous malaria field studies conducted between 2016 and 2021 in Djiba commune, this report emphasizes the burden of non-*falciparum* species, urging actions from the national malaria control program and decision-makers to implement specific control strategies.

## 2. Methodology

### 2.1. Study Site and Population

Data was gathered retrospectively using original research documents archived by the MRTC research team at the Catholic confessional center of Faladié. Faladje is the rural capital of the Ntjiba commune, located 80 km from the Malian capital Bamako. According to the local registers, the commune of Ntjiba has 23 villages with an estimated population of 23,000 inhabitants. Malaria is hyperendemic, and transmission is seasonal, mainly occurring between July and December when the rainfall is higher.

### 2.2. Data Sources

Epidemiological data collected consistently from October to December for six consecutive years (2016 to 2021) were compiled. The data consisting of microscopically confirmed malaria cases were extracted from registers. Patients visiting the health center clinic with malaria symptoms were designed as passive screening cases. At the community level, asymptomatic participants invited for weekly screening at the healthcare facility were classified as active screening individuals. The passive screening group comprises all malaria-positive patients of all ages visiting our health center while the active screening group included asymptomatic volunteers aged six months and more of the selected village site population invited for malaria screening.

Due to logistic impediments, there was no active screening from October to December 2016 and 2017, October 2018, November 2018, and October 2021. Blood smears were read independently by two microscopists, and any discrepancies were resolved by a third microscopist. Parasitemia and *Plasmodium* species identification were assessed as previously reported on thick and thin smears, respectively [18].

### 2.3. Study Data and Statistical Methods

For this study, the following parameters were considered: enrolment date (day, month, and year), participants’ age and sex, type of screening (passive or active), and clinical profile (symptomatic or asymptomatic). Parasitemia was estimated as parasite stage count by microliter (parasites/µL blood). Parasitemia was categorized as low (≤2500 parasites/µL blood), moderate (2501–199,999 parasites/µL blood), and high parasitemia (≥200,000 parasites/µL blood). Hemoglobin values less than 11.0 g/dL were classified as anemia and greater values as no anemia [19]. These categorical variables were presented as frequencies and percentages.

The data were recorded in Microsoft Office Access and analyzed with STATA software (version 15; Stata Corp., College Station, TX, USA). Pearson’s Chi-square test was used to compare *Plasmodium* spp. infections prevalence among screening groups, and participants’ clinical profile. The ANOVA test was used to compare variances across the different groups’ means (or averages). An alpha level of 0.05 was used for all tests to determine statistical significance.

### 2.4. Ethical Considerations

Our study used secondary data. The protocols used to populate the registers during primary data collection were all approved by the Ethics Committee of the Faculty of Medicine, Pharmacy and Odontology of the University of Bamako reference (Approval Codes: No 2O17/141/CE/FMPOS; No 2O19/168/CE/FMOS/FAPH; No 2O19/128/CE/FMPOS). The consent of the village communities was obtained from the local authorities before each study, and informed consent was obtained from all participants.

## 3. Results

### 3.1. General Characteristics Observed over the Years in the Studies Participants

A total of 8610 participants were enrolled in this study, 6870 from passive screening (symptomatic) and 1740 from active screening (asymptomatic). The number of participants infected with at least one plasmodial species were 65.20% (CI: 60.10–69.89%) and 22.41% (CI: 16.60–28.79%) in passive and active screening, respectively (Appendix A). As shown in Figure 1a, significatively, most participants were women throughout the study, with an overall male/female ratio of 0.81 (3858/4775) (*p* < 0.002). The median age of the participants was 12, ranging from 3 months to 85 years old. Children from 5 to 9 years old were the most represented, with 34.94% (CI: 28.89–39.67%) and 62.72% (CI: 61.69–89.97%) in passive and active screens, respectively (Figure 1b). The difference between age groups was statistically significant in passive (*p* = 0.027) and active (*p* < 0.001) screening.

### 3.2. Distribution of Malaria Prevalence by Gender and Age Groups

Overall, the prevalence of malaria was higher in males in both passive (69.12% vs. 62.24%, *p* = 0.003) and active screening (25.93% vs. 18.91%, *p* < 0.002) (Appendix A). Malaria prevalence appeared to be higher in males than females but the difference is only significant in passive screening (*p* = 0.034) (Figure 1c). Prevalence gradually decreased from 66.53% (95% CI: 57.60–75.41%) in 2016 to 49.53% (95% CI: 41.74–57.33%) in 2021 (Figure 1c, Appendix A).

In passive screening, the prevalence was high in the group of 5–9 (25.81%, 95% CI: 16.71–34.90%)), followed by 10–14 (18.65%, 95% CI: 14.56–22.74%) and 1–4 year olds (12.82%, 95% CI: 9.46–15.36%), (R^2^ = 0.9309, *p* < 0.004). The lowest prevalence was observed in children under one year (1.05%, 95% CI: 0.87–1.22%) and adults over 65 years (0.16%, 95% CI: 0.07–0.25%) (Appendix A).

As for the active screening, malaria infections were mostly affecting people aged from 5–9 (13.13%, 95% CI: 17.36–8.90%) and 10–14 year olds (7.32%, 95% CI: 4.09–10.55%). Only one child under one year old had parasites with no symptoms, while no asymptomatic infected individuals were detected beyond 25 years old (Figure 1d, Appendix A).

### 3.3. Plasmodium Sexual Stage Prevalence

In the passive screening group, the prevalence of *P. falciparum* gametocytes was 1.50% (95% CI: 0.80–2.20%) (*n* = 97) and 0.82% (95% CI: 0.47–1.16%) (*n* = 16) in children (5 to 9 years) and adolescents (10 and 14 years old), respectively. For the active screening group, *P. falciparum* gametocytes prevalence was 0.74% (95% CI: 0.43–0.97%) (*n* = 61) and 0.41% (95% CI: 0.13–0.68%) (*n* = 34) in 5- to 9-year-old children and 10- to 14-year-old adolescents, respectively. The lowest gametocyte carriage was found in children under one year of age (0.020%, 95% CI: 0.009–0.031%) and adults (25 and ≥65 years, (0.05%, 95% CI: 0.03–0.07%)) (Figure 1e). The overall carriage of gametocytes varied from month to month, depending on the year, with an average prevalence of 3.24% (95% CI: 1.63–5.77%) and 1.91% (95% CI: 0.43–0.97%) in passive and active screening, respectively (Appendix A). The means of gametocytaemia were similar with 132 (SD = 247) and 130 (SD = 245) gametocytes/µL of blood, respectively, in passive and active screening (*p* = 0.923).

### 3.4. Plasmodium Species Prevalence and Frequency

Monoinfections with *P. falciparum* represented the majority of malaria cases in symptomatic and asymptomatic individuals (Table 1). The prevalence and frequency among symptomatic individuals were 59.33% (95% CI: 48.35–70.27%) and 96.77% (95% CI: 93.46–100%), respectively. Similarly, in asymptomatic participants, the prevalence and frequency were 19.31% (95% CI: 15.43–23.19%) and 93.24% (95% CI: 89.26–97.40%), respectively (Table 1).

Interestingly, *P. malariae* was more frequent in asymptomatic participants (5.64%, 95% CI: 3.42–7.70%) than in participants with clinical symptoms (2.45%, 95% CI: 1.31–3.70%) (*p* < 0.001) (Table 1). On the other hand, *P. ovale* was the rarest species, with a prevalence and frequency under 1% regardless of the type of screening. The frequency of *P. ovale* varied between years and months (Table 1). No case of *P. vivax* was observed.

Coinfections of several *Plasmodium* species were also observed, mainly *P. falciparum + P. malariae (Pf + Pm)*. The frequency of *P. falciparum* slightly decreased consistently toward the month of December. In contrast, *P. malariae* was more frequent in December, corresponding to the end of the high-transmission season. The differences of *Plasmodium* species prevalence between age groups (R^2^ = 0.9737, *p* < 0.0001) and over years (R^2^ = 0.9309, *p* = 0.0040) were both statistically significant.

### 3.5. Non-Falciparum Malaria Species Are Associated with the Burden of Malaria Anemia

A higher number of carriers of *P. ovale* (50%,95% CI: 30–70%) and *P. malariae* (41.51%, 95% CI: 35.40–47.31%) carriers presented anemia (hemoglobin levels < 11 g/dL), while only 18.76% (95% CI: 14.93–22.53%) with infected with *P. falciparum* were anaemic. One of the five mixed *P. falciparum* + *P. malariae* infections had anemia (Table 1).

### 3.6. Plasmodium Species-Specific Characteristics

Parasitemia greater than 100,000 parasites/µL was only observed in *P. falciparum*, regardless of the type of screening (Figure 2). The average parasitemia for *P. falciparum* infections in symptomatic individuals was 37,372 (SD = 52,022). The average parasitemia was significantly lower in *P. malariae* 3120 parasites/µL (SD = 3092) and *P. ovale*, 3002 parasites/µL (SD = 2272) (R^2^ = 0.01448, *p* < 0.0001) (Figure 2a).

In asymptomatic participants, all species displayed low parasite loads as follow: *P. falciparum* with 10,535 parasites/µL (SD = 19,468) followed by *P. ovale* 5400 and finally *P. malariae* 1429 (SD = 1032) but the difference between the mean parasite densities is not statistically significant (R^2^ = 0.01453, *p* = 0.2643) (Figure 2b).

Concerning age groups, most participants between 25 and 64 years (88.83%, 95% CI: 79.92–97.71%) and ≥65 years (70.11%, 95% CI: 59.53–80.78%) had low parasite densities (1 to 2500 parasites/µL) (Appendix A). Participants under 24 years had moderate parasite densities (2501 to 199,999 parasites/µL). Hyper-parasitemia (≥200,000 parasites/µL of blood) was found only in symptomatic children under 10 years of age: 5–9 years (2.43%, 95% CI: 1.35–3.52%), 1–4 years (1.91%, 95% CI: 0.65–3.26%) and <1 year (0.10%, 95% CI: 0.07–0.17%) (Appendix A).

## 4. Discussion

The epidemiology of the various parasitic species of malaria found in Mali is poorly described. This study aimed to assess the epidemiological of plasmodial species in Ntjiba commune, in Mali, where malaria is hyperendemic and seasonal. All the data produced were obtained using conventional microscopy, which despite current advances in molecular diagnostic tools, remains the gold standard according to WHO [12]. Microscopic observation of blood smears collected across six different years in the high-transmission season, provided an accurate view of the prevalence and frequency of the different plasmodial species in this endemic area.

This study shows with clarity the high prevalence of clinical malaria in this period of each year corresponding to the peak of the transmission at the end of the rainy season, occurring between October and December in Mali. Our findings can be explained by the stability of malaria incidence at the high-transmission season and rainfall variability [20]. It has also been shown that the prevalence of symptomatic cases increases during periods of high transmission, while the number of asymptomatic carriers only increases in periods of low transmission [21]. Nevertheless, a slight progressive decrease, up to 5% lower, in the prevalence of clinical malaria was observed over the years, with an average prevalence of 65.20%. This decreasing prevalence is similar to that described in the 2018 summary report on the demographic and health survey in Mali [13] and results from permanent interventions of the National Malaria Control Programme (NMCP) to fight against this disease. The low number of screenings in the last two months of 2017 due to logistics issues may justify the drastic drop in prevalence that year. Although there was no active screening in certain periods, this would not impact our results, since we observed a stable prevalence of asymptomatic carriers across the whole duration of the study. These results are consistent with a previous multicentric study conducted in 2020 [22,23,24]. Our results do not show significant differences in the prevalence of *Plasmodium* species over the years. However, a change could occur with climatic vagaries since we observed an increase in *P. malariae* cases in asymptomatic people at the end of the high-transmission season. Regrettably, the number of malaria cases in the world suffered an increment in the last two years [1] mainly due to anti-COVID19 measures impeding implementing different preventive policies.

The high proportion of male participants infected in our data, even if there was a higher proportion of female subjects among the participants, could be explained due to sociocultural and behavioral factors that would expose them more to mosquito bites [25]; however, women are more vulnerable than men to malaria [26]. Additionally, along with the gradual reduction in clinical cases, a nonuniform distribution of malaria infections existed each year between age groups regardless the type of screening and a decrease in prevalence found in older participants. Children aged 5–9 years carried the greatest burden of malaria (25.81%) that was much higher than children under 5 years (12.82%) (Figure 1d). This fact could be explained by the effect of the Seasonal Malaria Chemoprevention (SMC) administered to younger children under 5 years during periods of high transmission [27,28]. Our findings are supported by previous studies from the same region that concluded that SMC significantly reduced the incidence of malaria [29,30]. However, the noncompliance with the SMC and the decrease with age of the protective effect of maternal antibodies could justify the slight prevalence in toddlers [31]. In fact, in 2018, only 60% of the cases had been confirmed in the country by optical microscopy, the current gold standard in the country [13], which raised concerns about the proper use of antimalarials.

Our results agree with a previous study carried out on malaria prevalence in a site with similar epidemiological characteristics in Mali [32], showing the presence of three plasmodial species among the four species existing in Mali, with *P. falciparum* being the most common (90% of the cases) regardless of the clinical profile of the participants. In our data, *P. falciparum* coinfections with *P. malariae* were more frequent than with *P. ovale*. Monoinfections with *P. malariae* were the second most frequent found in participants. Its prevalence and frequency around 2 and 5%, respectively, were seen to increase in asymptomatic people and in December, corresponding to the end of the high-transmission season. *P. ovale*, with a prevalence often below 1%, represented the third species of plasmodia found. Our data are consistent with a previous study carried out in this same locality [33] and with the results found in an entomological–epidemiological study on the transmission of malaria in Tibati, Cameroon [34]. In addition, we found no infections due to the species *P. vivax*. Although, microscopic evaluation of samples from some localities in the country suggested the presence of blood stages of *P. vivax*-like, later confirmed unambiguously by nested PCR [35]. African populations with the Duffy system antigen negative blood group were considered protected against *P. vivax* infection, but recent studies have shown cases of infections in Duffy-negative subjects in sub-Saharan countries [36].

Our analysis also shows us that *Plasmodium* species could be found in individuals of any age; however, adolescents aged from 10 to 14 years old were most frequently infected by non-*falciparum* species such as *P. malariae* and *P. ovale*. Malaria appeared to be more prevalent in children 5–9 and 10–14 years old, suggesting the importance of extending SMC to those age group. In particular, we found that only *P. falciparum* cases displayed hyperparasitemias, regardless of the type of screening, which are linked to severe malaria cases [37]. Furthermore, the results show that infections with non-*falciparum* species, as well as their mixed infections with *P. falciparum,* mostly showed moderate to low parasitemia [10]. None of the infections with *P. malariae* or *P. ovale* was responsible for severe malaria, as previously reported [38,39]. Parasite density clearly decreased with age. Children with malaria carried moderate parasite loads and from adulthood to the elderly, no participant presented severe symptoms due to hyper-parasitemia [40]. Studies have shown that people living in malaria-endemic areas gradually develop partial resistance to *P. falciparum* infection. However, protective immunity against non-*falciparum* species remained to be investigated as their biology differs from that of *P. falciparum* [41,42,43]. *P. malariae* and *P. ovale* infections have been shown to be responsible for anemia burden [44,45], lysis of erythrocytes, phagocytosis and sequestration of parasitized red blood cells and that explains the presence of anemia due to these species in our study [46]. Gametocytes, in *P. falciparum* infections, were found more in children aged 5 to 9 and the lowest carriage rate in those under one year and over 25 years of age, which was in agreement with previous studies [47] carried out in Bancoumana in Mali and in the same age group in Western Kenya [48].

A limitation to this study is that the recorded data only covered the end of the rainy season of each year from October to December, corresponding to a part of the high-transmission season. Therefore, the results do not provide the full epidemiological information for a whole year although malaria would be detectable even at periods of very low transmission between April and June (dry season). Longitudinal parasitological and entomological surveys covering a whole year for various years would be necessary to monitor the dynamics of malaria transmission in this malaria endemic area.

A second limitation is that the biological data are derived only from the results of microscopy, which may be the reason why no *P. vivax* infection has been detected in the population. It would therefore be necessary to combine microscopy with other tools, such qPCR, for the diagnosis of *plasmodium* species to closely monitor malaria infections and to assess the presence of *P. vivax,* because the study area has a cosmopolitan population comprising a strong community of Moors from northern Mali more likely to carry *P. vivax* [49], and support the data obtained for *P. malariae* and *P. ovale*.

As a third limitation, we found that there were few volunteers over 24 years of age who tested in the active screening compared to other age groups, representing a possible selection bias and a limitation of our study.

## 5. Conclusions

Malaria prevalence gradually decreases over the years, and parasite density of the different species varies depending on individuals’ clinical status (symptomatic/asymptomatic) and age. Non-*falciparum* species had low parasitemia. The frequency of *P. malariae* was higher at the end of high-transmission season (December) in asymptomatic carriers. *P. falciparum* was by far the most dominant species followed by *P. malariae*. *P. ovale* remained the third species regardless of the type of screening. Our characterization of *Plasmodium* species epidemiology highlights that *P. falciparum* represents a significant burden and that other *Plasmodium* species silently contribute to the malaria burden. If malaria elimination is to be achieved, awareness campaigns are needed to increase attention on *P. malariae* and *P. ovale* infections and patient care.

## Figures and Tables

**Figure 1 tropicalmed-08-00438-f001:**
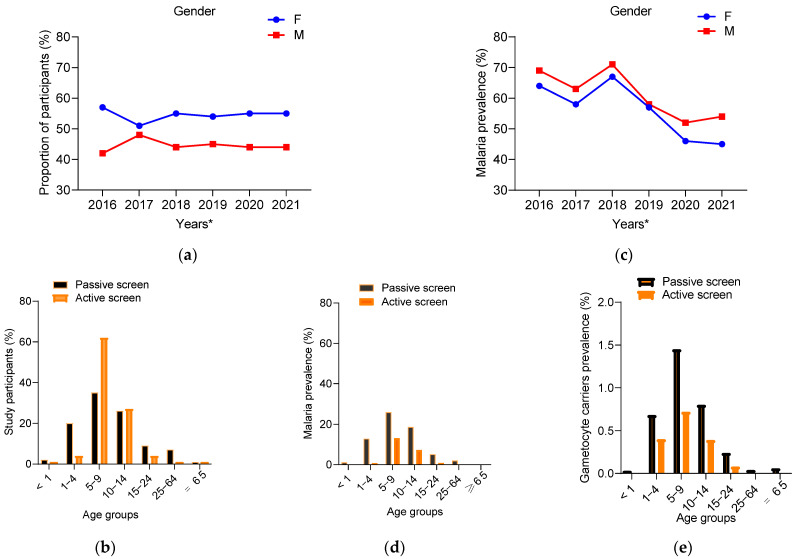
Distribution of study participants and malaria prevalence. (**a**) participants distribution by gender (F = Female, M = male); (**b**) distribution of passive and active screening participants by age groups; (**c**) malaria prevalence in each gender; (**d**) prevalence of parasite carriers detected in each age group; (**e**) prevalence of *P. falciparum* gametocyte carriers detected in symptomatic and asymptomatic participants distributed by age group.

**Figure 2 tropicalmed-08-00438-f002:**
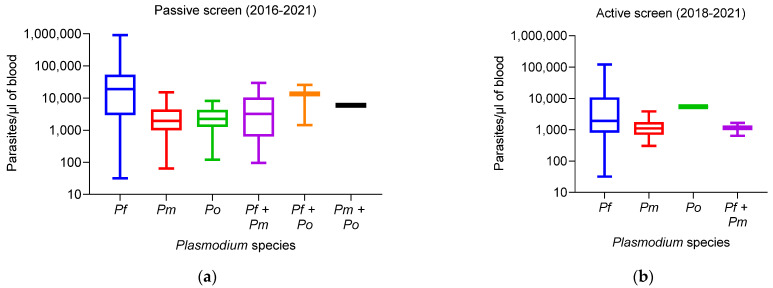
Parasitemia density detected in passive and active screens, The line inside each boxplot represents the median parasite densities per microliter of blood of each *Plasmodium* species. Boxplot representing the number of parasites count per microliter of blood for each *Plasmodium* species from (**a**) symptomatic patients (passive screening) and (**b**) asymptomatic parasite carrier participants (active screening). *Pf* = *Plasmodium falciparum*, *Pm* = *Plasmodium malariae*, *Po* = *Plasmodium ovale*.

**Table 1 tropicalmed-08-00438-t001:** Most *Plasmodium* species were detected during the three months of high transmission. Prevalence and frequency in passive and active screens, distribution stratified by age groups, per year and months, and anemia frequency are reported.

		**Plasmodial Species**					
**Parameters**		*Pf*	*Pm*	*Po*	*Pf + Pm*	*Pf + Po*	*Pm + Po*
Screen							
Passive	n1	4069	103	22	9	2	0
(N1 = 6858)	Prev	59.33	1.50	0.32	0.13	0.03	0.00
	Freq	96.77	2.45	0.52	0.21	0.05	0.00
Active	n2	336	20	1	2	0	1
(N2 = 1740)	Prev	19.31	1.15	0.06	0.11	0.00	0.06
	Freq	93.24	5.64	0.28	0.56	0.00	0.28
Age	Groups						
	<1	62 (1.40)	0	0	0	0	0
	1–4	774 (17.72)	7 (5.69)	1 (4.35)	1 (9.09)	0	0
	5–9	1831 (41.91)	34 (27.64)	5 (21.74)	2 (18.18)	1 (50)	0
	10–14	1254 (28.70)	65 (52.85)	13 (56.52)	7 (63.64)	1 (50)	1 (100)
	15–24	320 (7.30)	13 (10.57)	3 (13.04)	1 (9.09)	0	0
	25–64	122 (2.79)	3 (2.44)	1 (4.35)	0	0	0
	>65	8 (0.18)	1 (0.81)	0	0	0	0
Years	Months						
2016	Oct.	35 (97.22)	1 (2.78)	0	0	0	0
	Nov.	379 (95.23)	17 (4.27)	1 (0.25)	1 (0.25)	0	0
	Dec.	104 (90.43)	9 (7.83)	2 (1.74)	0	0	0
2017	Oct.	107 (92.24)	5 (4.31)	1 (0.86)	2 (1.72)	1 (0.86)	0
	Nov.	35 (97.22)	1 (2.78)	0	0	0	0
	Dec.	15 (88.24)	2 (11.76)	0	0	0	0
2018	Oct.	409 (97.61)	6 (1.43)	3 (0.72)	1 (0.24)	0	0
	Nov.	322 (97.87)	4 (1.22)	3 (0.91)	0	0	0
	Dec.	133 (95.68)	6 (4.32)	0	0	0	0
2019	Oct.	362 (99.18)	2 (0.55)	0	1 (0.27)	0	0
	Nov.	571 (97.61)	8 (1.37)	6	0	0	0
	Dec.	184 (93.88)	8 (4.08)	4 (2.04)	0	0	0
2020	Oct.	689 (96.50)	20 (2.80)	2 (0.28)	3 (0.42)	0	0
	Nov.	397 (96.13)	13 (3.15)	0	1 (0.24)	1 (0.24)	1 (0.24)
	Dec.	110 (91.67)	10 (8.33)	0	0	0	0
2021	Nov.	335 (98.24)	5 (1.47)	0	1 (0.29)	0	0
	Dec.	219 (96.48)	6 (2.64)	1 (0.44)	1 (0.44)	0	0
Total		4406 (96.50)	123 (2.69)	23 (0.50)	11 (0.24)	2 (0.04)	1 (0.02)
Anaemic	Status						
	anemia	6 (18.76)	44 (41.51)	5 (50.00)	1 (20.00)	0	0
	Normal	26 (81.24)	62 (58.49)	5 (50.00)	4 (80.00)	0	0

*Pf* = *Plasmodium falciparum*, *Pm* = *Plasmodium malariae*, *Po* = *Plasmodium ovale*, Prev = prevalence, Freq = frequence.

## Data Availability

The data presented in this study are available on request from the authors in an anonymized format upon reasonable request.

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
