# Peer review of "The Prevalence of Human Plasmodium Species during Peak Transmission Seasons from 2016 to 2021 in the Rural Commune of Ntjiba, Mali"

_tropicalmed, 2023, doi:10.3390/tropicalmed8090438_

Round 1
Reviewer 1 Report
Title and abstract
1. Based on the abstract, the title needs to be revised as "The prevalence of Plasmodium species during peak transmission seasons in the rural commune of Ntjiba, Mali" to more concise description of the manuscript content.
Introduction
2. In the first paragraph of the introduction, authors should talk about the impact of COVID-19 pandemic on malaria burden.
Method
3. The sentence "Blood samples were read 92 independently by two microscopists and any discrepancies were resolved by a third mi- 93 croscopist." Blood slide/smear should be appropiate.
4. Typo error...Plasmodium spp..should be spp.
5. Ethical considerations, any ethical number?
6. Line 149, please italic the genus of parasite.
Discussion
7. Line 258 Please discuss about why Mali did not have P. vivax case such as explain about Duffy antigen or genetic factors.
8. Line 269 the figures should not be cited in the discussion, please revise.
9. Line 272 authors stated that " no participant had severe malaria due to hyper-parasitaemia (33)" why? what is the threshold? how about immunity?
10. Line 297..Non-falciparum, Non-P. falciparum.
Referecesn
11. Poor reference format, authors must concentrate on reference format according the Journal style. Internet source should not be cited as reference.
Minor
Author Response
Please see the attachment in the box for a point-by-point response to the reviewer’s 1 comments.

Reviewer 2 Report
This is an interesting paper describing the prevalence of Plasmodium species over a period of 6 years in a malaria endemic region in Mali. The most interesting findings are the fact that P. malariae is highly prevalent in asymptomatic cases and falciparum in symptomatic cases. The study comprises a rather large study population.
There are a few points that require the attention of the authors.
First and most importantly is the use of English. Unfortunately, the manuscript contains several spelling errors, incorrect fonts (italics not adequately used for species) and wrong construction of sentences. I was trying to make some corrections on the manuscript, but the pdf file prohibited this.
The work is not actually looking at epidemiology. This would be much broader (including transmission dynamics, low malaria season, vectors etc.). It has a clear focus on prevalence. Therefore, wherever the manuscript is talking about epidemiology in the context of the manuscript it should be changed to prevalence – including in the title of the manuscript.
P. knowlesi is responsible for zoonotic infections in humans. This should be clarified.
The aim of the study should be better defined (last paragraph of the introduction/background). It is now more a short introduction to the Methods. What is the main objective? And why is it relevant.
It must be addressed in the discussion what the potential effect is of not having active screening during the three months of high transmission in 2016, 2017 and partly in 2018. Could this have affected prevalence of certain species etc.?
The remark on treatment (lines 97-98) is not relevant in the context of the presented work.
Supplementary figure 1e is mentioned twice in line 157 and is probably a mistake
Heading (line 178) should be changed, suggestion: Non-falciparum malaria species are associated with the burden of malaria anaemia (or something comparable – current heading does not make sense)
See comment above. Must be significantly improved
Author Response
Please see the attachment in the box for a point-by-point response to the reviewer’s 2 comments.

Round 2
Reviewer 1 Report
The manuscript is look better. I recommended authors to send the manuscript for English editing before acceptance. Reference lists must be edited carfullly before acceptance too.
I recommended authors to send the manuscript for English editing before acceptance
Author Response
Please see responses to comments in the attached PDF file
